# Postural and Proprioceptive Deficits Clinically Assessed in Children with Reading Disabilities: A Case-Control Study

**Franck Scheveig** [1,2] **and Maria Pia Bucci** [2,*]

1   Clinique de Posturologie, 66100 Perpignan, France
2   MoDyCo, UMR 7114 CNRS Université Paris Nanterre, 92000 Nanterre, France
*   Correspondence: mariapia.bucci@gmail.com

**Abstract:** Several studies have reported motor deficiencies in children with dyslexia, in line with the cerebellar deficit theory. In the present study, we explored whether tests used by physiotherapists during clinical evaluation were able to report motor deficits in a group of fifty-six dyslexic children (mean age $10.9 \pm 0.2$ years old) compared to a group of thirty-eight non-dyslexic children (mean age $11.2 \pm 0.4$ years old). The occurrence of instability on an unstable support; spinal instability in the sagittal, frontal and horizontal plane; head-eye discoordination; and poor eye stability were clinically assessed in the two groups of children. All such measures were found to be significantly more frequent in dyslexic than in non-dyslexic children ($p < 0.001$, $p < 0.05$, $p < 0.001$ and $p < 0.001$, respectively, for occurrence of instability on an unstable support, spinal instability, head-eye discoordination and poor eye stability). These results, firstly, confirmed the poor motor control of dyslexic children, suggesting deficient cerebellar integration. Secondly, for the first time, we reported that simple tests that can be done by pediatricians and/or during a clinical routine evaluation could be useful to discriminate children with reading difficulties. The tests used in this study could be a reference for a first exploration of motor deficiencies in children with dyslexia that can be easily assessed by clinicians and/or physiotherapists.

**Keywords:** reading disabilities; children; motor coordination; postural balance; proprioception

## 1. Introduction

In 1973, Frank and Levinson [1] were the first researchers to report poor stability in children with reading disabilities. They showed that the majority of children with reading disorders (97% of 115 children tested) had a positive Romberg test, difficulty in tandem walking, hypotonia, and various dysmetric or past-pointing disturbances during finger-to-nose and heel-to-toe tests, writing and drawing, as well as during ocular fixation and scanning testing. They hypothesized a cerebellar-vestibular impairment in children with dyslexia. Nicolson et al. [2] also showed postural instability and motor coordination deficits in children with dyslexia and suggested that a cerebellar deficit in dyslexia could cause such general motor impairments. Several subsequent studies by our group (i.e., [3,4] and other researchers [5–8] examining postural sway with a platform device in which children were asked to stand on a stable and/or unstable support showed significant instability in dyslexic children. Other studies reported poor motor control when dyslexic subjects had to perform dual tasks, for instance a cognitive and a motor task at the same time, most likely because of their incomplete development of automaticity [9,10]. All these studies supported the cerebellar deficit theory which posits that poor cerebellar development could be responsible for insufficient integration of the information necessary for automatized motor capabilities.

Note also that Martin da Cunhà [11,12] described a postural deficiency syndrome with a scoliotic body posture and vertebral/thoracic muscle hypotonia in children with dyslexia. He suggested that this pathology could be related to a deficit of the proprioceptive information system that is also controlled by the cerebellum.

Interestingly, for several years now, it has been suggested that the cerebellum plays a major role not only in motor activities but also in other cognitive tasks such as working memory, attention control and action planning (for a review, see [13]).

The cerebellum is well known to play a major role for the development of cortical network, particularly for the frontal and parietal cortex [14]; Stoodley and Schmahmann [15] reported also that the cerebellum is crucial for motor abilities but also for cognitive processes.

More recently, Shemesh and Zee [16] reported that different parts of the cerebellum were involved in the control of different types of eye movements (saccades, pursuits, gaze holding). It is well known that dyslexic children exhibit abnormal eye movements. For instance, Eden et al. [17] showed instability of eye fixation in 26 dyslexic children (11 years old) and Biscaldi et al. [18] reported more express saccades in dyslexic subjects compared to non-dyslexic subjects (from 12 to 32 years old), suggesting a deficit in attentional process in the dyslexic population. Our group [19] observed, in a large group of dyslexic children (from 7 to 14 years old), a significant difficulty in fixating on a visual target compared to non-dyslexic children of similar age. Such poor eye fixation capability could be due to an immaturity of central areas responsible for eye movement control.

Based on these findings, we examined further motor deficiencies clinically assessed in children with dyslexia. We used tests frequently employed by physiotherapists in order to explore possible motor deficiencies between a group of dyslexic children and a group of non-dyslexic children. All these tests were developed by physiotherapists who were not equipped with platforms, eye trackers or other systems allowing a precise measure of such motor deficiencies; consequently, their evaluation was based on the occurrence of instabilities and/or discoordination of the body and of segments of the body and instability of eye fixation.

## 2. Materials and Methods

### 2.1. Subjects

Ninety-four children participated in the study. The ELFE test (Évaluation de la Lecture en FluencE) (www.cogniciences.com, Grenoble, France, accessed on 26 August 2013) was used to select the children based on their reading ability, resulting in two groups: one group of thirty-eight children (mean age $11.1 \pm 0.4$ years old) with normal reading skills (G1) and another group of fifty-six children (mean age $10.9 \pm 0.2$ years old) with abnormal reading skills (G2). Note that the ELFE test is widely used by French laboratories/clinicians to evaluate the reading age of children. The test assesses whether reading abilities correspond to the chronological age of the child. In other words, for all children of G1 (normal readers), the reading age was similar to their chronological age, while for all children of G2 (dyslexic children), reading age was significantly lower than the chronological age; that is, their reading skill was abnormal with respect to their chronological age.

Dyslexic children were recruited after a neuropediatric assessment (done about 6 months before inclusion in the study) and non-dyslexic children were recruited from schools.

The Intelligence quotient (IQ) was also assessed using the Wechsler scale (Wechsler Intelligence Scale for Children, fourth edition [20]), by examining the verbal comprehension, perceptual reasoning, working memory and processing speed capabilities.

The inclusion criteria were no history of vestibular, orthopedic, neurological or psychiatric pathology; absence of drug use; normal or corrected-to-normal visual acuity (in each eye 20/25); normal mean intelligence quotient (IQ, evaluated with WISC-IV; between 80 and 115).

Exclusion criteria were any known neurological disorders, visual impairment, vestibular disorder, orthopedic disorder or surgeries and use of drugs.

The clinical characteristics of the two groups are shown in Table 1.

**Table 1.** Clinical characteristics of the two groups of children (G1 and G2) with mean and standard deviations of chronological age (years), height (cm) and weight (Kg), of the number of words read in one minute (ELFE test) and of the IQ score. Asterisks indicate that the value is significantly different ($p \leq 0.05$) between the two groups.

|  | G1 (n = 38) | G2 (n = 56) |
|---|---|---|
| Chronological age (years) | $11.1 \pm 0.4$ | $10.9 \pm 0.2$ |
| Height (cm) | $145 \pm 10$ | $149 \pm 15$ |
| Weight (Kg) | $45 \pm 7$ | $43 \pm 10$ |
| Number of words/min (ELFE test) | $139 \pm 5$ | $85 \pm 4$ * |
| IQ (WISC-IV) | $105 \pm 14$ | $100 \pm 12$ |

The investigation followed the principles of the Declaration of Helsinki and was approved by our Institutional Human Experimentation Committee (Comité de Protection des Personnes CPP Île-de-France). Written consent was obtained from the children's parents after the experimental procedure has been explained to them. Asterisk indicates that the value is significantly different ($p \leq 0.05$) between the two groups.

### 2.2. Evaluation Clinically Assessed

Stability on an unstable platform; spinal stability in the sagittal, frontal and horizontal plane of the head on the spine; head-eye coordination and eye stability were assessed by clinical tests performed by a physiotherapist. The total evaluation lasted about 30 min. In the following section, each of these tests will be described in more detail.

### 2.2.1. Stability on an Unstable Support

The child stands on an unstable support (BOSU, see Figure 1A) with his arms along his body, his legs straight and looks in front of him at a target (a cross of 2°) at a distance of 2 m. The oscillations are measured by an inclinometer (Clinometer Version 4.9.4 (2212183)) on IOS (Figure 1B). The inclinometer is a smartphone application for measuring inclinations in the sagittal and frontal plane. The child had to balance without moving for 10 s on the unstable support while fixating on the target, and overall postural oscillations greater than 5 degrees were measured (given that several clinical observations reported that normal body oscillation is between 1 and 4 degrees, see [21]).

(A)

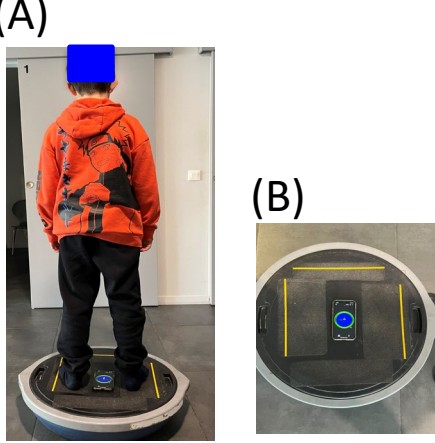

(B)

**Figure 1.** (**A**) The child stands on the BOSU support with his arms along his body, his legs straight and looks in front of him at a target (a cross of 2°) at a distance of 2 m. (**B**) Unstable support (BOSU support) with the inclinometer.

For each child, we measured the presence of postural oscillations greater than 5 degrees while the child was on the unstable platform (score 0 = oscillations greater than 5 degrees, score = 1 when oscillations greater than 5 degrees were present).

### 2.2.2. Spinal Stability in the Sagittal, Frontal and Horizontal Plane

The child stands facing a wall and is asked, firstly, to roll up with his chin to his chest and his legs stretched out; secondly, to stand up and look in front of him. The physiotherapist places a plumb line on the child's back next to the spine. The child presents a postural disorder in the sagittal plane if the plumb line is not in contact with vertebrae T6 and S2 (Figure 2A).

To measure the spinal alignment in the frontal plane, a laser is placed on the back of the child, projecting along the spinal axis. The distance between the line of the spine and the projection of the laser on the back is measured. The child presents a postural disorder in the frontal plane if the projection of the laser is not on the spinal line and the distance is greater than 2 cm (Figure 2B).

Finally, a protractor is used to measure the angle formed by the axis of the shoulders and the projection of the straight line formed by the edges of the heels in order to assess the postural deficit in the horizontal plane. The child presents a postural disorder in the horizontal plane if the projection of these two lines is not parallel (Figure 2C).

(A)            (B)            (C)

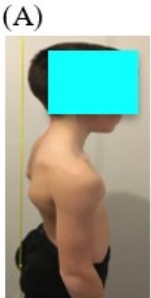 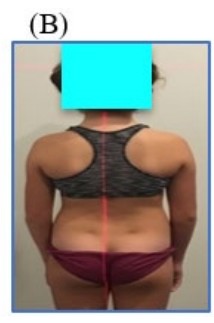 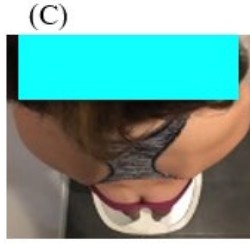

**Figure 2.** Measure of the spinal instability in the sagittal, frontal and horizontal plane. (**A**) Example of anterior scapular deviation in the sagittal plane. (**B**) Example of deviation in the frontal plane. (**C**) Example of right scapular torsion in the horizontal plane.

For each child, the total postural disorder in the sagittal or frontal or horizontal plane was evaluated by a score 0 or 1 if a disorder on one of these planes was present. If the child showed a disorder on 2 or 3 planes, the score given was 2 or 3.

### 2.2.3. Head-Eye Coordination and Eyes Stability Evaluation

The child is seated on an unstable rehabilitation chair (Satisform®, Saint-Saturnin, France), wearing a laser helmet (see Figure 3A) and is asked to follow horizontal, vertical and diagonal lines projected on the wall, placed in front of him at a distance of 2 m. The child must follow the lines back and forth as they are projected on the wall, using the laser helmet placed on his head during 3 min (see Figure 3B).

The variable measured is the stability of the head and trunk on the chair during such movements, and the head-eye coordination disorder is present if the laser beam does not stay on the lines and the laser movement is jerky when following the lines.

Lastly, the stability of the head with respect to the rest of the body is also measured by asking the child to fixate on a target (a cross of 2°) during at least 10 s (see Figure 3C).

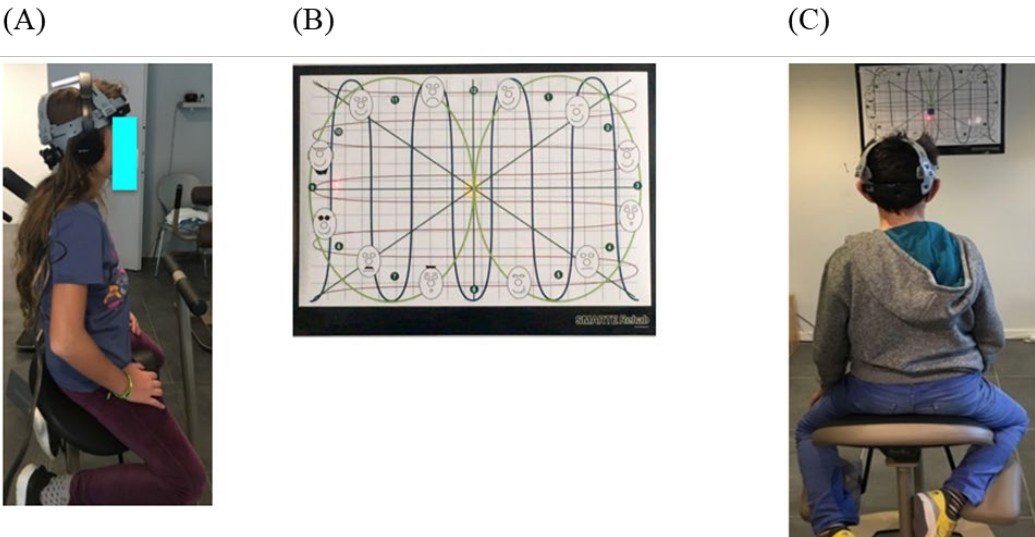

**Figure 3.** Evaluation of head-eye coordination and eye stability. (**A**) The child is seated on an unstable rehabilitation chair (Satisform®) and is wearing a laser helmet; (**B**) he has to follow horizontal, vertical and diagonal lines projected on the wall placed in front of him at a distance of 2 m. (**C**) The child is seated on an unstable rehabilitation chair (Satisform), is wearing a laser helmet and he has to fixate on a target 2 m in front of him.

For each child in group G1 and G2, a score = 1 was given when head-eye discoordination was present and a score = 1 was also given when the eye fixation was not maintained for 10 s.

*2.3. Statistical Analysis*

As the Shapiro–Wilk test demonstrated that the data were not normally distributed, and all the statistics were non-parametric, the non-parametric Mann–Whitney *U*-test was used to compare the age, the number of words read in one minute (ELFE test) and the postural scores recorded in the two groups of children (G1 and G2). In order to evaluate the strength of a statistical claim, we calculated the effect size for all variables given by the rank biserial correlation; the level of significance was kept at 0.05. All statistical analyses were processed using JASP software (a free and open-source program for statistical analysis supported by the University of Amsterdam).

**3. Results**

Firstly, we compared cognitive characteristics in the two groups of children included in the study. The Mann–Whitney *U*-test showed a significant difference only for the number of words read in one minute (ELFE test, $p < 0.001$, effect size = 0.75, see Table 1). In other words, in G1, the children showed normal reading speed, while in G2, the children had a significantly abnormal reading slowness.

Table 2 shows, respectively, the occurrence of instability when the child was on an unstable support, the occurrence of spinal instability in different planes, the occurrence of head-eye discoordination and the occurrence of eye fixation instability measured in the two groups of children.

For instance, the instability on unstable support was observed in 9 of the 38 children examined in G1 and in 41 of the 56 children in G2. The Mann–Whitney *U*-test showed that instability on an unstable platform was significantly more frequently observed in children with reading disabilities (G2) than in children with normal reading skills (G1) ($p < 0.001$, effect size = 0.49). The occurrence of spinal instability in different planes was also reported in the two groups of children tested. The group of children with reading disabilities (G2) showed significantly more instability (54 children of the 56 total tested) than the group of

children with normal reading capabilities, G1 (only in 17 of the 38 children tested, $p < 0.001$, effect size = 0.50).

**Table 2.** Occurrence of instability on an unstable platform, of spinal instability in different planes, of head-eye discoordination and of eye fixation instability reported in the children of the two groups of normal and poor readers, G1 and G2, respectively). Asterisks indicate that the value is significantly different ($p \leq 0.05$) between the two groups.

|  | G1 (n = 38) | G2 (n = 56) |
|---|---|---|
| Instability on unstable platform | 9/38 | 41/56 * |
| Spinal instability in different planes | 17/38 | 54/56 * |
| Head-eye discoordination | 8/38 | 44/56 * |
| Eye fixation instability | 13/38 | 52/56 * |

Head-eye discoordination occurred more frequently in children with reading disorders (in 48 of the 56 children tested in G2) than in normal readers (only 8 of the 38 children, $p < 0.001$, effect size = 0.57). Lastly, the quality of eye fixation was significantly poor in children with reading disabilities (reported in 52 of 56 children tested in G2) with respect to normal readers (only 13 of 38 children in G1, $p < 0.001$, effect size = 0.59).

## 4. Discussion

The present study reported that spinal instability, head-eye discoordination and poor eye fixation occurred significantly more frequently in dyslexic children that in non-dyslexic children. The novelty of the study is that such motor deficiencies can be easily measured by using simple tests that could be widely employed during the screening of children; for instance, during a pediatric visit or a routine evaluation in order to make a rapid preliminary evaluation of dyslexia, allowing a rapid and simple diagnosis. All these tests can be easily used by clinicians to detect motor control deficiencies that need to be taken into account by specific reeducation programs.

Interestingly, in a recent study, Marchetti et al. [22] applied the M-ABC test to a group of dyslexic and non-dyslexic subjects (from 18 to 29 years old). Note that this test is widely used in clinical practice in France to assess motor deficiencies in children with neurodevelopmental deficits [23]. The authors observed that sensorimotor impairment occurred more frequently in the group of dyslexics (27%) than in the non-dyslexic group (5% only) and they hypothesized that this percentage might be even higher if these subjects underwent a complete evaluation of postural capabilities. Furthermore, they reported that the group of dyslexics with sensorimotor impairment showed an impaired internal representation of action and greater variability in execution durations when performing a mental imagery paradigm. This finding is in line with a previous study [5] reporting that variability in motor parameters could suggest poor motor control, also in line with a more recent study [24] reporting that poor cognitive mechanisms controlling mental sensorimotor representation in dyslexic subjects could be responsible for the slowness of actual as well as mental movements in dyslexic subjects.

Beckinghausen and Sillitoe [25] reviewed cerebellar connectivity and highlighted the role of the cerebellum in posture, coordination and motor activities. Additionally, Stoodley [26] and Carreiras et al. [27] showed, during reading activities, the relationship between cerebellar regions (VI and VII lobules) and the cerebral regions of the left inferior frontal lobe and the left inferior occipitotemporal cortex. Stoodley [26] reported reduced gray matter in specific cerebellar structures and impairment in the cerebral cortical network in dyslexic subjects and in subjects with neuro-developmental pathologies. Taken together, all these studies are in line with the hypothesis of a cerebellar deficit put forward by Nicolson et al. [2] in dyslexia subjects.

Further imaging studies will be necessary to precisely localize the cerebellar structures responsible for motor instability in the dyslexic population and follow up studies will be useful to explore possible adaptive mechanisms able to reduce this motor deficit. Importantly, it is well known that the cerebellum exhibits adaptive mechanisms; indeed, there are some studies showing an improvement in postural stability and cognitive skills in dyslexics. Reynolds et al. [28] reported beneficial effects of exercise-based treatment on balance, dexterity and eye movement control. Gouleme et al. [29] reported an improvement in postural control after a short visuo-postural training period, and more recently, Ramezani et al. [30] obtained improvements in both balance and cognitive skills after a dual task training in which both verbal working memory and motor abilities were trained.

Further studies on a larger number of children with both normal and abnormal reading skills will be needed in order to better develop simple tests for clinicians to evaluate the impact of dyslexia on motor control. Imaging studies combined with motor tests in children with dyslexia will be necessary in order to confirm the hypothesis of a cerebellar deficiency in dyslexia.

## 5. Conclusions

This study reported for the first time, motor deficits in dyslexic children by employing a battery of simple tests that can be used during routine clinical evaluation by pediatricians to screen children with dyslexia easily. According to previous findings, such motor and coordination deficits could be due to poor cerebellar activity, supporting the cerebellar deficit hypothesis in dyslexia.

**Author Contributions:** F.S. curated and analyzed the data and wrote the manuscript. M.P.B. supervised, reviewed and edited the manuscript. All authors have read and agreed to the published version of the manuscript.

**Funding:** This research received no external funding.

**Institutional Review Board Statement:** The study was conducted in accordance with the Declaration of Helsinki and approved by the Institutional Human Experimentation Committee (CEEI-IRB IRB00003888) (Code: 16-290/Date: 5 April 2016).

**Informed Consent Statement:** Informed consent was obtained from all subjects involved in the study. Written informed consent has been obtained from the patients to publish this paper.

**Data Availability Statement:** The data presented in this study are available on request from the corresponding author.

**Acknowledgments:** The authors thank the children who participated in this study.

**Conflicts of Interest:** The authors declare that the research was conducted in the absence of any commercial or financial relationships that could be construed as a potential conflict of interest.

## Abbreviations

| | |
|---|---|
| ELFE test | Évaluation de la Lecture en FluencE |
| IQ | Intelligence Quotient |

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
