# Peer review of "Postural and Proprioceptive Deficits Clinically Assessed in Children with Reading Disabilities: A Case-Control Study"

_2411-5150, 2023_

Round 1

Reviewer 1 Report (Previous Reviewer 1)

The authors addressed all my comments. I am satisfied with how the paper looks.

Abstract - lines 13 - 17 - the sentence is not readable. I don't know what the p-values refer to. Please reword it.

Material and Methods:

Subjects -Please complete the information on the height and weight of the persons included in the study.

Results -Table 2 should be described better. The values in the table are like 9/38. It should be clarified what 9 and 38 mean. i.e. G1 (.... / ....).

Author Response

see doc attached

Reviewer 2 Report (Previous Reviewer 3)

Review comments:

Well done on putting together a comprehensive research study. In the manuscript, the authors substantially show the presentation of the contributions and the experimental results of their proposed issues. In my opinion, the version of the manuscript is now acceptable for publication.

Author Response

thank you for your comments

This manuscript is a resubmission of an earlier submission. The following is a list of the peer review reports and author responses from that submission.

Round 1

Reviewer 1 Report

All notes and comments are in the appendix. Please try to do correlations of dyslexia test results with physio test results. I'm curious if there will be any correlations.

Author Response

see doc

Reviewer 2 Report

I have revised this manuscript with attention. I have adversited some concerns that may be solved and other red flags that can difficult the publication of this manuscript.

In the title, it would be recommendable to add the type of design used. For example, maybe "a case-control study"?

In abstract must appears the most important statistics data with p-value.

I would add more keywords, such as proprioception and postural control. The correct MeSH for balance is "postural balance".

In methods, where is the ethics permission? Ethics permission is mandatory for studies involving human research. 

Inclusion and exclusion criteria are not mentioned.

Characteristics of the participants must be explained in results (first subsection)

Please, add abbreviations in tables and figures.

Discussion is shorter but well planned.

Author Response

see doc

Reviewer 3 Report

Review comments:

This manuscript entitled “Postural and proprioceptive deficits clinically assessed in children with reading disabilities” aimed to examine further motor deficiencies clinically assessed in children with dyslexia.

Although this study was read with interest, more work needs to be done in the manuscript to show clearly the potential of the study. In the Introduction Section, how the postural and proprioceptive deficits are important to the children with reading disabilities? It is unclear. On the other hand, in the Materials and Methods Section, why choose the unstable platform as the clinically assessed evaluation?

Author Response

see doc

Round 2

Reviewer 2 Report

Ethical considerations have not been attended.